# Seroprevalence and risk factors for hepatitis B and hepatitis C in three large regions of Kazakhstan

Alexander Nersesov[1,2‡], Arnur Gusmanov[3‡], Byron Crape[3], Gulnara Junusbekova[2,4], Salim Berkinbayev[1,2], Almagul Jumabayeva[1,2], Jamilya Kaibullayeva[1,2], Saltanat Madenova[2], Mariya Novitskaya[2], Margarita Nazarova[2], Abduzhappar Gaipov[3], Aiymkul Ashimkhanova[3], Kainar Kadyrzhanuly[3], Kuralay Atageldiyeva[3], Sandro Vento[5], Alpamys Issanov[3] *

1 Asfendiyarov Kazakh National Medical University, Almaty, Kazakhstan, 2 Research Institute of Cardiology and Internal Diseases, Almaty, Kazakhstan, 3 Department of Medicine, Nazarbayev University School of Medicine, Nur-Sultan, Kazakhstan, 4 Kazakh Medical University of Continuing Education, Almaty, Kazakhstan, 5 Faculty of Medicine, University of Puthisastra, Phnom Penh, Cambodia

‡ AN and AG should be considered joint first authors.
* alpamys.issanov@nu.edu.kz

**Data Availability Statement:** All relevant data are within the paper and its Supporting Information files.

## Abstract

### Background & aims

Kazakhstan has implemented comprehensive programs to reduce the incidence of Hepatitis B and Hepatitis C. This study aims to assess seroprevalence and risk factors for HBsAg and anti-HCV positivity in three large regions of Kazakhstan.

### Methods

A cross-sectional study was conducted in three regions geographically remote from each other. Participants were randomly selected using a two-stage stratified cluster sampling and were surveyed by a questionnaire based on the WHO STEP survey instrument. Blood samples were collected for HBsAg and anti-HCV testing.

### Results

A total of 4,620 participants were enrolled. The seroprevalence was 5.5% (95%CI: 3.6%-8.4%) for HBsAg and 5.1% (95%CI: 3.5%-7.5%) for anti-HCV antibodies. Both were more prevalent in the western and northern regions than in the southern. A history of blood transfusion was significantly associated with anti-HCV presence, with odds ratios (ORs) of 2.10 (95%CI: 1.37–3.21) and was borderline associated with HBsAg 1.39 (95%CI: 0.92–2.10), respectively. Having a family member with viral hepatitis was also borderline associated (2.09 (95%CI: 0.97–4.50)) with anti-HCV positivity.

### Conclusions

This study found a high-intermediate level of endemicity for HBsAg and a high level of endemicity for anti-HCV antibodies in three large regions of Kazakhstan. We found that history of surgery was not associated with HbsAg neither with anti-HCV seropositivity rates.

**Funding:** SB received a grant # 48973/PCF-MON-OT-17 from Ministry of Education and Science of the Republic of Kazakhstan (https://www.gov.kz/memleket/entities/edu?lang=kk). AG received the funding (Funder Project Reference: 240919FD3913) from the Nazarbayev University Faculty Development Research Grant Program FDCRGP 2020-2022 (https://nu.edu.kz/). The funders had no role in study design, data collection and analysis, decision to publish, or preparation of the manuscript.

**Competing interests:** The authors have declared that no competing interests exist.

Blood transfusion was associated with anti-HCV seropositivity, however, to investigate effectiveness of the introduced comprehensive preventive measures in health care settings, there is a need to conduct further epidemiological studies.

## Introduction

Viral hepatitis is a major public health threat and one of the leading causes of mortality and disability worldwide, with a number of related deaths similar to HIV, malaria and tuberculosis. [1] Chronic hepatitis B virus (HBV) and hepatitis C virus (HCV) infections constitute over 90% of the overall burden of viral hepatitis. In 2015, approximately 257 million and 71 million people worldwide had chronic hepatitis B and hepatitis C infections respectively, predominantly in developing countries, and were responsible for an estimated 1.3 million deaths. [2] HBV and HCV infections also decrease the quality of life and pose significant financial burdens on patients and their families, including health-related expenses and adverse impacts on employment. [3, 4]

Kazakhstan, an upper-middle income country, is the largest country in Central Asia, and has a population of around 18 million, sparsely distributed throughout the vast country. Slightly less than half of the population lives in rural areas, lacks access to adequate healthcare and has poorer health indicators than people living in urban areas. [5] Studies evaluating the epidemiology of HBV and HCV in Kazakhstan have included only populations-at-risk, and/or were conducted in a single city or region. [6–13] In the last decade, hepatological services have been improved, and policies and interventions to prevent viral hepatitis implemented, including a vaccination campaign against HBV, screening of blood donors and populations-at-risk for HBV and HCV, and establishment of national clinical treatment guidelines. [14] However, the impact of these actions on HBV and HCV–associated risk factors remains unclear. Understanding the epidemiology of HBV and HCV can inform more effective policies to decrease the burden of viral hepatitis in Kazakhstan that may be applicable to other developing countries as well.

Our study is the first geographical-diverse, randomized study investigating the prevalence of HBsAg and anti-HCV antibodies in three large regions of Kazakhstan.

## Materials and methods

### Study design

This study is an extension of a national cross-sectional study on the monitoring of non-communicable diseases (NCD) supported by the Ministry of Health, which utilized the standardized WHO STEPwise approach to NCD surveillance (STEPS). [15]

A two-stage stratified cluster sampling was utilized. Given that Kazakhstan consists of 14 oblasts (administrative districts), three oblasts with larger populations–Almaty (size ~ 4 mln) in the South, Pavlodar (size ~ 0.8 mln) in the North, and Aktobe (size ~ 0.9 mln) in the West–were selected. In the first stage, each oblast was stratified into large cities, small towns and villages, according to population size, as primary sampling units (PSU). 35 PSUs were randomly selected across all strata. In the second stage, within each randomly selected stratum, 100 households were selected using systemic random sampling technique from a list of households in each selected PSU. Trained nurse interviewers collected data by face-to-face interviews from randomly selected households between January 2015 and December 2017.

## Data collection

In the first step of data collection, participants were surveyed about their socio-demographic, behavioral risk factors and history of NCD using the WHO STEPS survey instrument. [16] The WHO STEPS survey instrument was developed by leading experts in their respective fields and used for national risk factor surveillance in many countries worldwide. [17] The instrument was translated by bilingual experienced researchers to Russian language. After experts fluent in both languages assessed appropriateness of the translation, the instrument was back translated to English by an independent translator to check for appropriateness of the translation to the original version. Later, the instrument was pre-tested among 50 participants and was revised for clarity and wording.

In the second step, physical measurements such as height, weight, blood pressure and other anthropometric measurements were collected by trained nurse interviewers. In the last step, after completing the survey and collecting anthropometric measurements, participants were invited to their local outpatient clinics for venous blood sample collection. An experienced phlebotomist collected venous blood specimen into Vacutainer serum separator tubes. The serum was separated from cells within 2 hours of collection and stored at 4˚C temperature (not more than 48 hours) until assayed. Serum samples were tested for hepatitis B surface antigen (HBsAg), and anti-HCV antibody (anti-HCV) using Elecsys HBsAg II and Elecsys Anti-HCV (Roche Diagnostics GmbH, Germany) enzyme-linked immunosorbent assays (ELISA) according to the manufacturer's instructions on the cobas e 601 platform. All results were reported as positive or negative. The instruments were calibrated daily based on standardized procedures.

A total of 4,620 participants responded to the survey. Out of these participants, 3,694 for HBsAg and 3,697 for anti-HCV were included in the study. Participants were excluded because of indeterminate serological test results (due to hemolysis or borderline results) or incomplete data (S1 Appendix).

The study was conducted in accordance with the Declaration of Helsinki and received ethical approval from the Ethics Board of the National Institute of Cardiology and Internal Diseases (Protocol #18 dated 28.01.2015) and the Institutional Research Ethics Committee of Nazarbayev University (NUSOM-IREC-NOV-2019-#20). Eligible participants provided written consent.

## Data variables

Using the WHO STEPS survey instrument, data on socio-demographic characteristics and behavioral risk factors were collected. Socio-demographic variables included age categories (18–29, 30–39, 40–59, and 60–88 years of age), educational status (none/high school level, vocational level and university level), ethnicity (Kazakh; Russian; and others (Uzbek, which comprises 1% of the population; Ukrainian, 4.2%; other Asians, 4.5%; other European ethnicities, 1.8%), type of residence (urban or rural) and region (South, North and West). Any settlement with more than 50,000 people was considered an urban area, while any settlement with at least 50 residents and not exceeding 50,000 people was defined as a rural area. [18] Participants were asked questions concerning potential risk factors for infection, including history of any surgery, blood transfusion, hemodialysis, having a family member with viral hepatitis, injecting drug usage and tattooing.

## Statistical analysis

After conducting basic descriptive statistics for all participants (means, medians, standard deviations, frequencies), bivariate analysis was performed to compare differences between

independent variables and HBsAg or anti-HCV status. In bivariate analysis between socio-demographic and outcome variables, the prevalence of viral hepatitis with 95% confidence intervals and p-values were reported. Chi-square or Fisher's exact tests for categorical variables and the Student's T-test or Mann-Whitney U test for continuous independent variables were conducted, as appropriate. Existence of spatial autocorrelations in HBsAg and anti-HCV sero-positivity by the region were tested using the Global Moran's I test. Given spatial autocorrelations were identified (Moran's I = 0.006 with p<0.001 for HBsAg and Moran's I = 0.005 with p<0.001 for anti-HCV), multivariable mixed effects logistic regression was used to account for clustering of cases within the region and potential non-independence of the outcomes. In these models, associations between risk factors and the prevalence of outcomes as determined by HBsAg and anti-HCV results were presented as adjusted odds ratios while controlling for age and sex. Final models were built through inclusion of: first, demographic variables (age, sex and ethnicity), then additionally two potential risk factors (history of having surgery and history of blood transfusion), and later, all other risk factors (family member with viral hepatitis, hemodialysis and having tattoo or piercing). Given that we used two-stage cluster sampling, all descriptive, bivariate and multivariable statistics were weighted using inverse probability weighting to adjust for sampling design and non-response rate. All analyses were conducted using STATA 15 statistical software. [19]

## Results

### Seroprevalence and risk factors for HBV

Among the 3,694 respondents, 218 individuals, which corresponded to 5.5% (95%CI: 3.6%-8.4%) weighted prevalence, were identified as HBsAg-seropositive. The highest prevalence of HBsAg was observed among the young age group 30–39 years old (Fig 1). In bivariate analysis, the weighted prevalence of HBsAg among males at 6.5% (95%CI: 4.7–8.9), was borderline statistically significantly higher than females, who had a weighted prevalence of 5.1% (95%CI:

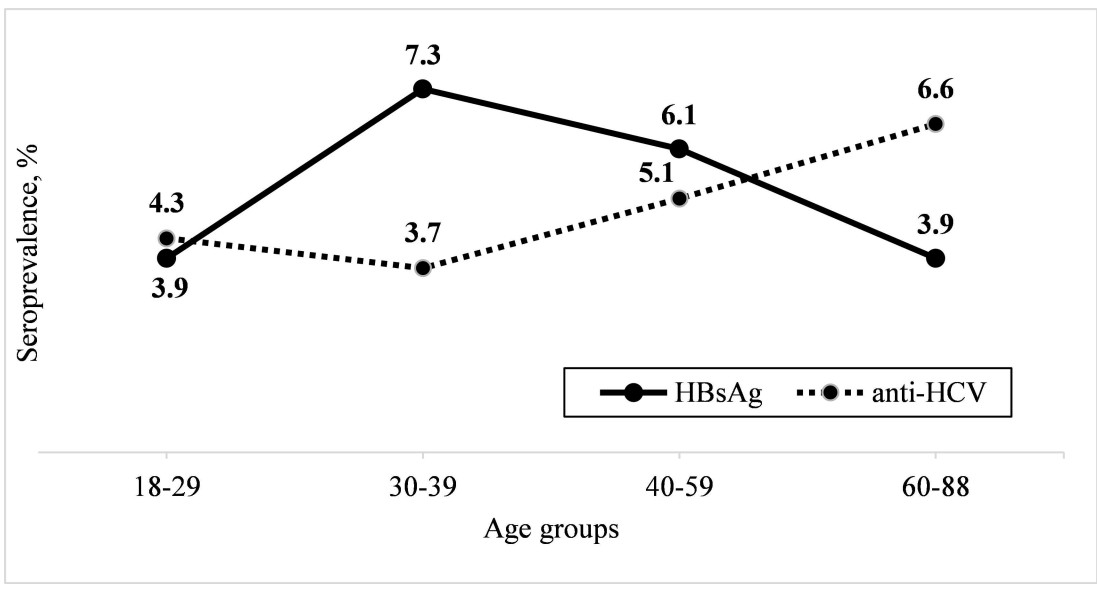

**Fig 1. Age-specific distributions of HBsAg and anti-HCV seroprevalence.**

**Table 1. Bivariate analysis for associations of socio-demographics characteristics with prevalence of HBsAg seropositivity.**

| Characteristics | Number of tested individuals | Unweighted prevalence of HBsAg % (95% CI) | Weighted prevalence of HBsAg % (95% CI) | p-value[†] |
|---|---|---|---|---|
| All participants | 3,694 | 5.9 (5.2–6.7) | 5.5 (3.6–8.4) | - |
| Age categorical | | | | 0.12 |
| 18–29 | 555 | 4.5 (2.9–6.6) | 3.9 (2.0–7.6) | |
| 30–39 | 513 | 7.8 (5.6–10.5) | 7.3 (4.2–12.4) | |
| 40–59 | 1,762 | 6.1 (5.0–7.3) | 6.1 (3.9–9.3) | |
| 60–88 | 864 | 5.3 (3.9–7.0) | 4.0 (2.0–7.7) | |
| Gender | | | | 0.07 |
| Females | 2,827 | 5.5 (4.7–6.4) | 5.1 (3.1–8.3) | |
| Males | 867 | 7.3 (5.6–9.2) | 6.5 (4.7–8.9) | |
| Ethnicity | | | | 0.08 |
| Kazakh | 2,482 | 6.3 (5.4–7.3) | 5.9 (0.4–8.8) | |
| Russian | 767 | 4.6 (3.2–6.3) | 2.8 (1.1–7.1) | |
| Other ethnicities | 417 | 5.5 (3.5–8.2) | 5.4 (3.5–8.2) | |
| Education level | | | | 0.75 |
| None/school level | 827 | 5.3 (3.9–7.1) | 5.9 (2.7–12.4) | |
| Vocational level | 1,754 | 5.9 (4.9–7.1) | 5.4 (3.8–7.6) | |
| University level | 1,104 | 6.2 (4.8–7.7) | 5.1 (3.2–8.2) | |
| Residence | | | | 0.35 |
| Urban | 2,106 | 5.9 (4.8–7.2) | 4.0 (1.4–10.3) | |
| Rural | 1,588 | 5.9 (4.9–7.0) | 6.3 (4.6–8.5) | |
| Region | | | | 0.02 |
| South | 749 | 2.4 (1.4–3.8) | 2.7 (1.4–5.1) | |
| West | 1,491 | 6.8 (5.6–8.2) | 7.3 (6.1–8.6) | |
| North | 1,454 | 6.7 (5.5–8.1) | 7.0 (3.7–13.0) | |

[†]Estimated by Chi-square test or Fisher's exact test.

Number (percent) of participants missing answer for ethnicity– 28 (0.8%); missing answer for education– 9 (0.2%).

3.1%-8.3%) (Table 1). HBsAg seropositivity of 7.3% (95%CI: 6.1%-8.6%) and 7.0% (95%CI: 3.7%-13.0%) in West and North Kazakhstan respectively, were statistically significantly higher than the 2.7% (95% CI: 1.4%-5.1%) in the South. No difference in the seroprevalence was observed between urban and rural areas (Fig 2A).

In Table 2, a weighted multivariable mixed effects logistic regression analysis adjusting for age and sex, showed that having a reported history of blood transfusion increased the odds of having a HBsAg positive result by 34% as compared to those not reporting any blood transfusion (p-value = 0.02). Those reporting to have tattoo or piercing had lower odds of having HBsAg when compared to those without (adjOR = 0.44, 95%CI: 0.27–0.72, p-value = 0.001).

In the final weighted multivariable mixed effects logistic regression analysis (Table 3) with HBsAg status as the outcome, three variables were found to be statistically significant: age, sex, and having tattoo or piercing. Those who were males (adjOR = 1.5; 95%CI: 1.31–1.72), in 30–39 age group, in 30–39 age group (adjOR = 1.98; 95%CI: 1.21–3.25) or 40–59 age group (adjOR = 1.64; 95%CI: 1.19–2.26), were more likely to be HBsAg-positive. However, those who had tattoo or piercing were less likely to have HBsAg positive result (adjOR = 0.45; 95% CI: 0.29–0.7). Also, in the Model 2, those reporting a history of blood transfusion were more likely to have HBsAg seropositivity with borderline significance (adjOR = 1.38; 95%CI: 0.97–1.95; p = 0.07).

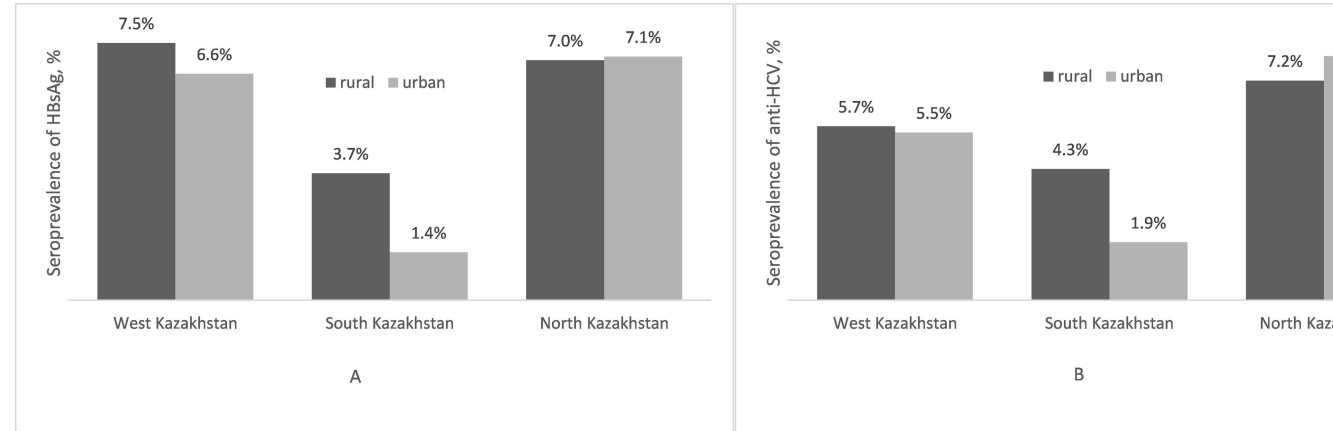

**Fig 2.** Regional and residential differences in the seroprevalence of HBsAg (a) and anti-HCV antibodies (b).

## Seroprevalence and risk factors for HCV

A total of 222 out of 3,697 participants, which accounted for weighted prevalence of 5.1% (95% CI: 3.5%-7.5%), were found to be seropositive for anti-HCV (Table 4). Anti-HCV seropositivity had a tendency to increase with older ages (Fig 1). The 3.1% prevalence of anti-HCV in South Kazakhstan (95%CI: 1.8%-5.3%) was lower than the prevalence in West and North Kazakhstan at 5.7% (95%CI: 3.2%-9.9%) and 7.5% (95%CI: 5.2%-10.7%), respectively, with a p-value = 0.07. No difference in the seroprevalence was observed between urban and rural areas (Fig 2B).

As shown in Table 5, a weighted multivariable mixed effects logistic regression analysis adjusting for age and sex showed an 114% increase in odds of having anti-HCV positive results for participants who reportedly had family members who were infected with viral hepatitis as

**Table 2. Risk factors for hepatitis B: Bivariate analysis and multivariable mixed effects logistic regression analysis adjusting for age and sex.**

| Variables | Number tested | Weighted prevalence of HBsAg % (95% CI) | p-value[†] | Adjusted OR[‡] (95%CI) | p-value for adjusted OR |
|---|---|---|---|---|---|
| Family member having viral hepatitis | | | 0.93 | | 0.57 |
| **No** | 3,383 | 5.5 (3.6–8.3) | | 1.0 | |
| Yes | 275 | 5.3 (2.3–11.9) | | 0.90 (0.63–1.29) | |
| History of surgery | | | 0.84 | | 0.93 |
| **No** | 1,963 | 5.4 (3.5–8.2) | | 1.0 | |
| Yes | 1,696 | 5.6 (3.3–9.4) | | 1.02 (0.64–1.62) | |
| Having tattoo or piercing | | | 0.29 | | 0.001 |
| **No** | 3,299 | 5.7 (3.6–9.0) | | 1.0 | |
| Yes | 352 | 3.2 (1.2–8.5) | | 0.44 (0.27–0.72) | |
| Having hemodialysis | | | 0.25 | | 0.32 |
| **No** | 3,634 | 5.5 (3.6–8.4) | | 1.0 | |
| Yes | 18 | 1.6 (0.1–16.4) | | 0.39 (0.06–2.46) | |
| History of blood transfusion | | | 0.34 | | 0.02 |
| **No** | 3,154 | 5.3 (3.2–8.4) | | 1.0 | |
| Yes | 506 | 6.8 (4.1–11.3) | | 1.34 (1.04–1.74) | |

[†]Chi-square test or Fisher's exact test as appropriate.

[‡]Models were adjusted for age and sex.

**Table 3. Final multivariable mixed effects logistic regression models for HbsAg seropositivity.**

| Variables | HBsAg seropositivity | | HBsAg seropositivity | | HBsAg seropositivity | |
|---|---|---|---|---|---|---|
| | OR (95% CI) | p-value | OR (95% CI) | p-value | OR (95% CI) | p-value |
| | Model 1 | | Model 2 | | Model 3 | |
| Categorical age | | 0.03 | | 0.03 | | 0.03 |
| *18–29* | Ref. | | Ref. | | Ref. | |
| *30–39* | 1.94 (1.20–3.14) | | 1.92 (1.20–3.08) | | 1.98 (1.21–3.25) | |
| *40–59* | 1.62 (1.09–2.41) | | 1.58 (1.12–2.24) | | 1.64 (1.19–2.26) | |
| *60–88* | 1.00 (0.47–2.11) | | 0.99 (0.51–1.91) | | 1.02 (0.54–1.92) | |
| Sex | | <0.001 | | <0.001 | | <0.001 |
| *Female* | Ref. | | Ref. | | Ref. | |
| *Male* | 1.39 (1.19–1.62) | | 1.41 (1.21–1.64) | | 1.50 (1.31–1.72) | |
| Ethnicity | | 0.21 | | 0.23 | | 0.26 |
| *Kazakh* | Ref. | | Ref. | | Ref. | |
| *Russian* | 0.48 (0.16–1.42) | | 0.46 (0.16–1.32) | | 0.50 (0.16–1.53) | |
| *Other ethnicities* | 1.25 (0.77–2.05) | | 1.26 (0.74–2.15) | | 1.28 (0.76–2.14) | |
| History of having surgery | - | - | | 0.92 | | 0.99 |
| *No* | | | Ref | | Ref. | |
| *Yes* | | | 0.98 (0.60–1.59) | | 0.99 (0.62–1.61) | |
| History of blood transfusion | - | - | | 0.07 | | 0.11 |
| *No* | | | Ref. | | Ref. | |
| *Yes* | | | 1.38 (0.97–1.95) | | 1.39 (0.92–2.10) | |
| Family member having viral hepatitis | - | - | - | - | | 0.71 |
| *No* | | | | | Ref. | |
| *Yes* | | | | | 0.91 (0.56–1.48) | |
| Having hemodialysis | - | - | - | - | | 0.18 |
| *No* | | | | | Ref. | |
| *Yes* | | | | | 0.38 (0.09–1.59) | |
| Having tattoo or piercing | - | - | - | - | | <0.001 |
| *No* | | | | | Ref. | |
| *Yes* | | | | | 0.45 (0.29–0.70) | |

Model 1 included age, sex and ethnicity. Model 2 = Model 1+ history of having surgery and history of blood transfusion. Model 3 = Model 2 + family member having viral hepatitis, history of hemodialysis and having tattoo or piercing.

compared to those who had not (p-value = 0.03). Having had a blood transfusion also increased the odds of having anti-HCV antibodies by 117% (p-value<0.001).

In the final weighted multivariable mixed effects logistic regression analysis (Model 6, Table 6), blood transfusion was statistically significantly associated with HCV seropositivity, while having family member with hepatitis showed borderline significance. Those who had a family member infected with viral hepatitis (adjOR = 2.09; 95%CI: 0.97–4.5), who had a history of blood transfusion (adjOR = 2.1; 95%CI: 1.37–3.21) were more likely to be anti-HCV positive.

Models differed in that age and sex were significantly associated only in the model with HBsAg, and history of previous blood transfusion only in the model with anti-HCV.

## Discussion

We have investigated the seroprevalence and associated risk factors of HBsAg and anti-HCV antibodies in the three large regions of Kazakhstan utilizing randomized data. The prevalence

**Table 4. Socio-demographics: Prevalence of anti-HCV seropositivity, bivariate analysis.**

| Characteristics | Number tested | Unweighted prevalence of HCV % (95% CI) | Weighted prevalence of anti-HCV % (95% CI) | p-value[†] |
|---|---|---|---|---|
| All participants | 3,697 | 6.0 (5.3–6.8) | 5.1 (3.5–7.5) | - |
| Age categorical | | | | 0.55 |
| 18–29 | 553 | 5.4 (3.7–7.6) | 4.3 (1.1–14.9) | |
| 30–39 | 510 | 6.1 (4.2–8.5) | 3.7 (1.6–8.4) | |
| 40–59 | 1,762 | 5.8 (4.7–7.0) | 5.1 (3.8–6.7) | |
| 60–88 | 872 | 6.8 (5.2–8.6) | 6.6 (4.0–10.8) | |
| Gender | | | | 0.78 |
| Females | 2,827 | 5.5 (4.7–6.5) | 5.2 (3.5–7.6) | |
| Males | 870 | 7.5 (5.8–9.4) | 4.9 (2.9–8.3) | |
| Ethnicity | | | | 0.52 |
| Kazakh | 2,480 | 6.9 (5.0–6.9) | 4.9 (3.4–7.2) | |
| Russian | 771 | 5.8 (4.3–7.7) | 4.9 (2.6–8.9) | |
| Other ethnicities | 418 | 6.9 (4.7–9.8) | 6.3 (3.4–11.4) | |
| Education level | | | | 0.49 |
| None/school level | 828 | 6.2 (4.6–8.0) | 5.8 (4.2–8.1) | |
| Vocational level | 1,754 | 5.5 (4.5–6.7) | 4.7 (2.9–7.7) | |
| University level | 1,106 | 6.6 (5.2–8.2) | 5.1 (3.3–7.9) | |
| Residence | | | | 0.49 |
| Urban | 2,102 | 6.0 (5.0–7.1) | 5.6 (3.8–8.2) | |
| Rural | 1,595 | 6.0 (5.0–7.3) | 4.2 (1.8–9.1) | |
| Region | | | | 0.07 |
| South | 750 | 2.9 (1.8–4.4) | 3.1 (1.8–5.3) | |
| West | 1,488 | 6.0 (4.8–7.3) | 5.7 (3.2–9.9) | |
| North | 1,459 | 7.6 (6.3–9.1) | 7.5 (5.2–10.7) | |

[†]Chi-square test or Fisher's exact test as appropriate.

Number (percent) of participants missing answer for ethnicity– 28 (0.8%); missing answer for education– 9 (0.2%).

of HBsAg and anti-HCV antibodies among the study population was found to be 5.5% and 5.1%, respectively, both much higher than the prevalence for the WHO European Region (0.9% for HBV and 1.1% for HCV infections). [20]

Previously, studies showed lower prevalence of HCV in Kazakhstan. [21, 22] A recent systematic review estimated a prevalence of anti-HCV antibodies at 0.7% for the general population of Kazakhstan; however the studies included in the review were largely sampling non-randomly among blood donors and were limited to single locations, potentially introducing important bias. [22] The meta-analyses conducted by Gower et al. throughout different countries in Central Asia estimated a 3.3% population had anti-HCV antibodies in Kazakhstan, [21] excluding studies in blood donors. Though their computed prevalence estimate for Kazakhstan was lower that the prevalence found in our study, the country was still found to have the second highest HCV seroprevalence in Central Asia. [21] In different studies, the prevalence of HBsAg seropositivity in Kazakhstan was found to be similar to other countries in the Central Asia region, falling into the range of high-intermediate level of endemicity. [23]

Numerous studies have established the effectiveness of vaccination in reducing HBV infection. [23, 24] In our study, the seroprevalence of HBsAg was low in the young age group, then sharply increased among 30–39 years old, and decreased in older ages. We can speculate that the low prevalence of HBsAg among the young age group might be attributed to the introduction of universal HBV vaccination in 1998; [14] the vaccination covered more than 97% of

**Table 5. Risk factors for anti-HCV antibodies: Bivariate analysis and multivariable logistic regression analysis adjusting for age and sex.**

| Variables | Number tested | Weighted prevalence of HCV % (95% CI) | p-value[†] | Adjusted OR[‡] (95% CI) | p-value for adjusted OR |
|---|---|---|---|---|---|
| Injection drug use | | | 0.85 | | 0.74 |
| No | 3,639 | 5.1 (3.5–7.4) | | 1.0 | |
| Yes | 22 | 5.9 (0.7–34.7) | | 1.49 (0.14–16.04) | |
| Family member having viral hepatitis | | | <0.01 | | 0.03 |
| No | 3,388 | 4.7 (3.2–6.9) | | 1.0 | |
| Yes | 274 | 10.2 (5.4–18.4) | | 2.14 (1.09–4.21) | |
| History of surgery | | | 0.25 | | 0.43 |
| No | 1,963 | 4.6 (2.8–7.4) | | 1.0 | |
| Yes | 1,700 | 5.7 (3.9–8.4) | | 1.22 (0.75–1.98) | |
| Having tattoo or piercing | | | 0.22 | | 0.17 |
| No | 3,301 | 4.8 (3.1–7.5) | | 1.0 | |
| Yes | 354 | 7.8 (4.0–14.4) | | 1.73 (0.79–3.78) | |
| Having hemodialysis | | | 0.28 | | 0.48 |
| No | 3,638 | 5.1 (3.5–7.5) | | 1.0 | |
| Yes | 18 | 1.6 (0.1–16.9) | | 0.36 (0.20–6.30) | |
| History of blood transfusion | | | <0.001 | | 0.001 |
| No | 3,158 | 4.4 (2.9–6.8) | | 1.0 | |
| Yes | 506 | 9.0 (5.7–14.0) | | 2.17 (1.40–3.35) | |

[†]Chi-square test or Fisher's exact test as appropriate.

[‡]Models were adjusted for age and sex.

children born after 1995. However, perinatal transmission studies and early childhood HBV prevalence studies are needed to confirm the effectiveness of the vaccination campaign. The presence of the highest number of infections in the 30–39 years of age group suggests that HBV transmission happened in a recent past (10–30 years) [25] and might have been due to sexual, horizontal or vertical transmission, before the universal vaccination was introduced.

The incidence of HBV and HCV infections in Western countries has been declining since screening of donated blood for blood borne infections, safer injection practices and universal HBV vaccination in infants were introduced. [26] Currently, in those countries, horizontal transmission among adult at-risk populations, such as IVDUs and people with high-risk sexual contacts, has become the most common. In low and middle-income countries, there is growing evidence that health-related procedures are the main route of transmission of HBV and HCV. [26, 27] Inconsistent screening of blood donors and unsafe injection practices both by professionals and nonprofessionals are common means of transmission.

Surgical procedures are recognized as an important route of transmission for HBV infection worldwide. [28] Our findings could not find associations of a history of surgery with HBV nor with HCV infection. The introduction of compulsory vaccination of all medical workers and routine surveillance for HCV and HBV infections among most of them, especially those working in surgical departments, may have played a role in reducing the risk of transmission from medical workers to patients. [29] However, failure to comply with infection safety measures could contribute to iatrogenic transmission in the future. Thus, there is a need to conduct further studies and strengthen the hospital-associated infections surveillance system in Kazakhstan, as data is limited. [30]

A blood transfusion history was associated with increases in the anti-HCV seropositivity rate while it was a borderline significant with HBsAg positivity. In low and middle income

**Table 6. Final multivariable mixed-effects logistic regression models for anti-HCV seropositivity.**

| Variables | Anti-HCV seropositivity | | Anti-HCV seropositivity | | Anti-HCV seropositivity | |
|---|---|---|---|---|---|---|
| | OR (95% CI) | p-value | OR (95% CI) | p-value | OR (95% CI) | p-value |
| | Model 4 | | Model 5 | | Model 6 | |
| Categorical age | | 0.70 | | 0.88 | | 0.92 |
| *18–29* | Ref. | | Ref. | | Ref. | |
| *30–39* | 0.80 (0.15–4.14) | | 0.72 (0.13–4.06) | | 0.71 (0.13–4.00) | |
| *40–59* | 1.10 (0.60–2.02) | | 0.96 (0.50–1.82) | | 0.91 (0.49–1.70) | |
| *60–88* | 1.52 (0.69–3.32) | | 1.35 (0.55–3.32) | | 1.38 (0.54–3.53) | |
| Gender | | 0.64 | | 0.76 | | 0.55 |
| *Female* | Ref. | | Ref. | | Ref. | |
| *Male* | 1.11 (0.85–1.46) | | 1.05 (0.77–1.43) | | 1.13 (0.75–1.70) | |
| Ethnicity | | 0.56 | | 0.46 | | 0.47 |
| *Kazakh* | Ref. | | Ref. | | Ref. | |
| *Russian* | 0.85 (0.35–2.08) | | 0.82 (0.30–2.24) | | 0.80 (0.28–2.28) | |
| *Other ethnicities* | 1.39 (0.91–2.13) | | 1.50 (0.95–2.36) | | 1.48 (0.95–2.31) | |
| History of blood transfusion | | - | | <0.001 | | 0.001 |
| *No* | | | Ref | | Ref. | |
| *Yes* | | | 2.21 (1.60–3.05) | | 2.10 (1.37–3.21) | |
| History of surgery | - | - | | 0.83 | | 0.94 |
| *No* | | | Ref. | | Ref. | |
| *Yes* | | | 1.06 (0.64–1.75) | | 1.02 (0.65–1.60) | |
| Injection drug use | - | - | - | - | | 0.99 |
| *No* | | | | | Ref. | |
| *Yes* | | | | | 1.01 (0.05–21.27) | |
| Family member having viral hepatitis | - | - | - | - | | 0.06 |
| *No* | | | | | Ref. | |
| *Yes* | | | | | 2.09 (0.97–4.50) | |
| Having tattoo or piercing | - | - | - | - | | 0.24 |
| *No* | | | | | Ref. | |
| *Yes* | | | | | 1.70 (0.71–4.10) | |
| Having hemodialysis | - | - | - | - | | 0.20 |
| *No* | | | | | Ref. | |
| *Yes* | | | | | 0.23 (0.02–2.17) | |

Model 4 included age, sex and ethnicity. Model 5 = Model 4+ history of having surgery and history of blood transfusion. Model 6 = Model 5 + family member having viral hepatitis, history of hemodialysis and having tattoo or piercing.

countries, the transmission of viral hepatitis through blood transfusion remains a critical problem. [2] Fifteen years ago it was estimated that 31 out 142 developing countries did not screen donated blood for blood-borne infections, and additional 37 did not screen regularly. [27] In Kazakhstan, blood donors at high-risk for having HBV and HCV are screened through the use of a questionnaire before donation and serological testing. In 2009, health authorities mandated testing of all donated blood for anti-HCV antibodies, anti-HIV antibodies and HBsAg. Thus, to reduce risk of window period infection transmission, a four-month storage of plasma was also introduced in 2009, for which at the end of the four months the donor is tested again before the usage of donated plasma. [31] During the years 2012–2015, a second stage of testing, i.e. Polymerase Chain Reaction (PCR) for donated blood was added to increase the sensitivity and reduce the risk of transmission of infections. [32] Even though we did not collect data on

dates of blood transfusion, we can speculate that infected participants who received donated blood may have had transfusions before the more rigorous blood bank screening programs were put in place. In any case, the high prevalence of anti-HCV antibodies in all age groups and their increase with age (Fig 1) suggests higher exposure to the infection in a recent (10–30 years) and distant past (30–50 years). [25]

We also found an association between having a family member with any viral hepatitis and the presence of anti-HCV antibodies. A systematic review conducted by Waure et al. indicates that household members are at risk of HCV infection if someone in that household is infected. [33] There are no definitive explanations for this association. However, some studies have suggested that transmission could occur through using the same razors, scissors or toothbrushes among family members. [34]

We found no differences in the prevalence of HBsAg and anti-HCV antibodies between urban and rural areas. The highest prevalence of both was observed in West and North Kazakhstan; this finding may be associated with regional inequities in healthcare, health financing and health outcomes, [5] given that the highest quality health services are provided in the South.

## Strengths and limitations

This study has several strengths. The large sample size allowed for higher statistical power to measure prevalence and risk factors with greater precision. The survey was conducted in three large regions, remotely located from each other, increasing potential national representativeness of the findings. In addition, a randomized multistage cluster sampling method was utilized, which could increase generalizability of the findings. Finally, HBsAg and anti-HCV antibodies were determined using serological tests rather than self-reporting.

However, several limitations of the study must be considered. Independent risk factors were obtained using self-reported information, including having tattoo or piercing, which could be biased. Also, important potential risk factors, such as occupation (e.g. healthcare worker), sexual behaviors and more in-depth questions for other risk factors were not included in the survey. Additionally, anti-HCV antibodies can simply indicate past rather than current infection, and HCV-RNA should have been tested to determine the number of current HCV-infected individuals. Lastly, the study respondents from three regions may not completely represent the general population of Kazakhstan since sex (overrepresented by female participants) and age (the majority were in age group of 40–88 years old) distributions were not comparable to the national demographical statistics.

## Conclusions

In conclusion, the study estimates the seroprevalence of HBsAg and anti-HCV antibodies in the three large regions of Kazakhstan, highlighting the high-intermediate and high levels of endemicity, respectively. We found that history of surgery was not associated with HBsAg, neither with anti-HCV seropositivity rates. Blood transfusion was associated with anti-HCV seropositivity, however, to investigate effectiveness of the introduced comprehensive preventive measures in health care settings, there is a need to conduct further epidemiological studies.

## Supporting information

**S1 Appendix. Flowchart of inclusion and exclusion of participants.**
(PDF)

**S2 Appendix. Global Moran's I test for spatial autocorrelation of HBsAg and anti-HCV seropositivity by region.**
(PDF)

**S3 Appendix. Russian translated the WHO STEPS instrument.**
(PDF)

**S4 Appendix. The study dataset in excel format.**
(XLS)

**S5 Appendix. STROBE checklist.**
(PDF)

## Acknowledgments

The authors thank staff of the Research Institute of Cardiology and Internal Diseases, Almaty, Kazakhstan for helping in data collection.

## Author Contributions

**Conceptualization:** Alexander Nersesov, Arnur Gusmanov, Byron Crape, Abduzhappar Gaipov, Aiymkul Ashimkhanova, Kainar Kadyrzhanuly, Kuralay Atageldiyeva, Sandro Vento, Alpamys Issanov.

**Data curation:** Alexander Nersesov, Gulnara Junusbekova, Salim Berkinbayev, Almagul Jumabayeva, Jamilya Kaibullayeva, Saltanat Madenova, Margarita Nazarova, Kuralay Atageldiyeva, Alpamys Issanov.

**Formal analysis:** Arnur Gusmanov, Byron Crape, Abduzhappar Gaipov, Kainar Kadyrzhanuly, Alpamys Issanov.

**Investigation:** Alexander Nersesov, Arnur Gusmanov, Gulnara Junusbekova, Salim Berkinbayev, Almagul Jumabayeva, Jamilya Kaibullayeva, Saltanat Madenova, Mariya Novitskaya, Margarita Nazarova, Kuralay Atageldiyeva.

**Methodology:** Alexander Nersesov, Gulnara Junusbekova, Salim Berkinbayev, Mariya Novitskaya, Abduzhappar Gaipov, Aiymkul Ashimkhanova, Kainar Kadyrzhanuly, Kuralay Atageldiyeva, Sandro Vento, Alpamys Issanov.

**Project administration:** Alexander Nersesov, Gulnara Junusbekova, Salim Berkinbayev, Almagul Jumabayeva, Jamilya Kaibullayeva, Saltanat Madenova, Mariya Novitskaya, Margarita Nazarova, Kuralay Atageldiyeva.

**Resources:** Salim Berkinbayev, Almagul Jumabayeva, Jamilya Kaibullayeva, Saltanat Madenova, Mariya Novitskaya, Margarita Nazarova, Abduzhappar Gaipov, Kuralay Atageldiyeva, Alpamys Issanov.

**Software:** Arnur Gusmanov, Alpamys Issanov.

**Supervision:** Alexander Nersesov, Byron Crape, Gulnara Junusbekova, Salim Berkinbayev, Saltanat Madenova, Mariya Novitskaya, Margarita Nazarova, Alpamys Issanov.

**Validation:** Arnur Gusmanov, Saltanat Madenova, Aiymkul Ashimkhanova, Alpamys Issanov.

**Visualization:** Arnur Gusmanov, Alpamys Issanov.

**Writing – original draft:** Alexander Nersesov, Arnur Gusmanov, Aiymkul Ashimkhanova, Sandro Vento, Alpamys Issanov.

**Writing – review & editing:** Arnur Gusmanov, Byron Crape, Abduzhappar Gaipov, Aiymkul Ashimkhanova, Kainar Kadyrzhanuly, Kuralay Atageldiyeva, Sandro Vento, Alpamys Issanov.

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
