## [Decision Letter · Decision Letter 0]

18 Feb 2021

PONE-D-20-31835

Seroprevalence and risk factors for hepatitis B and hepatitis C in the general population of Kazakhstan

PLOS ONE

Dear Dr. Issanov,

Thank you for submitting your manuscript to PLOS ONE. After careful consideration, we feel that it has merit but does not fully meet PLOS ONE’s publication criteria as it currently stands. Therefore, we invite you to submit a revised version of the manuscript that addresses the points raised during the review process.Indded, there are several issues that must be addressed prior to consideration for publication, first of all, the notion of "Nationwide" is elusive in this case, methotological aspects may be improved and therefore the discussion section.

We look forward to receiving your revised manuscript.

Kind regards,

Isabelle Chemin, PhD

Academic Editor

PLOS ONE

Journal Requirements:

2)  Please include additional information regarding the survey or questionnaire used in the study and ensure that you have provided sufficient details that others could replicate the analyses. For instance, if you developed a questionnaire as part of this study and it is not under a copyright more restrictive than CC-BY, please include a copy, in both the original language and English, as Supporting Information, or include a citation if it has been published previously.

3) In the Methods, please discuss whether and how the questionnaire was validated and/or pre-tested. If these did not occur, please provide the rationale for not doing so.

4) In the discussions about your research, please take care to avoid statements implying causality from correlational research, e.g. “iatrogenic factors seem to continue to play a significant role as a route of transmission”. A cross-sectional study can be used to explore associations and not causality due to a number of limitations such a failure to determine temporal precedence.

5) In statistical methods, please clarify whether you corrected for multiple comparisons.

6) In your statistical analyses, please state whether you accounted for survey weights and clustering by region.

7) As part of your revision, please complete and submit a copy of the STROBE checklist, a document that aims to improve reporting and reproducibility of observational studies for purposes of post-publication data analysis and reproducibility: (http://www.strobe-statement.org). Please include your completed checklist as a Supporting Information file. Note that if your paper is accepted for publication, this checklist will be published as part of your article.

8)  We note that the grant information you provided in the ‘Funding Information’ and ‘Financial Disclosure’ sections do not match.

9)  We note that you have indicated that data from this study are available upon request. PLOS only allows data to be available upon request if there are legal or ethical restrictions on sharing data publicly. For information on unacceptable data access restrictions, please see http://journals.plos.org/plosone/s/data-availability#loc-unacceptable-data-access-restrictions.

Reviewers' comments:

Reviewer's Responses to Questions

**Comments to the Author**

1. Is the manuscript technically sound, and do the data support the conclusions?

Reviewer #1: Partly

2. Has the statistical analysis been performed appropriately and rigorously? 

Reviewer #1: No

3. Have the authors made all data underlying the findings in their manuscript fully available?

Reviewer #1: No

4. Is the manuscript presented in an intelligible fashion and written in standard English?

Reviewer #1: Yes

5. Review Comments to the Author

Reviewer #1: Overall this is an interesting paper that describes an important disease globally and in Kazakhstan. The paper estimated hepatitis B and hepatitis C for a representative sample of the Kazakhstan population for three large oblasts (provinces). The goal was to relate seroprevalence estimates to several demographic or sociological factors. While important work, there are several major methodological issues that need to be addressed before consideration for publication.

ABSTRACT: there are abbreviations for the viruses used in the abstract that need to be spelled out like the authors did in the first paragraph of the introduction.

INTRODUCTION: I'm concerned about calling this a national study when only three oblasts were used to represent a landscape as largest as Kazakhstan. Having been there and worked there for several years, I appreciate that these three areas represent a large proportion of the population. However, I do not think the authors have done enough to describe the distribution of people or describe the proportion of people outside of these three areas and how they may or may not differ from those inside of the study area. I'm not sure it is fair to call this national as designed or at least described.

METHODS: there are several issues that must be addressed prior to consideration for publication.

First, as a serological surveillance study I would expect to read a much more clear methodology of how samples were collected, processed, and the serological tests performed. Overall, the methods section lacks appropriate citations and detailed enough methodology to repeat this work. Without expanding on both citations in detail this work should not be considered reproducible.

In the discussion of the data variables for this study, the authors provide very little information on how urban and rural were defined. These terms are specifically defined in a variety of studies using different cutoffs for population density. Likewise, across these three regions of Kazakhstan, there is a growing periurban environment where rural migrants are moving closer to the city and the city suburbs are expanding into the rural area. This study at a minimum needs to define urban and rural.

The larger issue with this study is the use of backward stepwise variable selection for the regression modeling. It has been well known for many years now that stepwise selection techniques are losing favor and there is a large body of statistical literature describing why. It has become a best practice to use a multi-model approach, for example, looking at all possible variable combinations with something like a dredge approach. Those models would then be evaluated with something like an AIC. Equally concerning is the use of the spatial location, at least defined by region, as a covariate without first testing for spatial autocorrelation. Each of these three regions was selected because they represent a large proportion of the human population in Kazakhstan. Human populations by their nature are typically clustered in space meaning their covariates are likely autocorrelated. The authors here did nothing to test for spatial autocorrelation either in the input data or in the residuals of the models. Without such tests is difficult to assess whether or not the data for this study meet the assumptions of independence for the regression model selected. Likewise, it has become more usual to see a mixed model approach where a random effect would be used for region. These issues should be addressed and should not simply be addressed by adding one or two sentences to the methodology, but rather performing additional model evaluation, selection, and spatial autocorrelation testing.

Building on concerns about the regression model, the authors also used the T-test for the variables without first testing for normality. It may be more likely that a nonparametric test touches the Mann-Whitney U test may be more appropriate to analyze the same question.

Later in the results the authors state there were some adjustments for the models and those were not provided.

DISCUSSION: the discussion of the paper is well written. However the discussion highlights the results of the study, which do not address the concerns raised above about the methodology performed. It is important to ensure that the statistical analysis was appropriate before pacing and discussion that supports those results.

I think this is an interesting paper and addresses a very serious disease concern. The methodological comments I raise here can be addressed and should strengthen the paper.

6. PLOS authors have the option to publish the peer review history of their article (what does this mean?). If published, this will include your full peer review and any attached files.

Reviewer #1: No

---

## [Author Response · Author response to Decision Letter 0]

2 May 2021

Comments and Suggestions for Authors

1) Thank you for submitting your manuscript to PLOS ONE. After careful consideration, we feel that it has merit but does not fully meet PLOS ONE’s publication criteria as it currently stands. Therefore, we invite you to submit a revised version of the manuscript that addresses the points raised during the review process.Indded, there are several issues that must be addressed prior to consideration for publication, first of all, the notion of "Nationwide" is elusive in this case, methotological aspects may be improved and therefore the discussion section.

Response:

We thank the Editor for this comment. We agree that the study respondents from three regions may not completely represent the general population of Kazakhstan since sex (overrepresented by female participants) and age (the majority were in age group of 40-88 years old) distributions were not comparable to the national demographical statistics. Thus, we modified accordingly the text and the title of the manuscript. We also included more details to the methodology, recalculated estimates using a different statistical approach and improved the discussion section based on the changes in the results.

Response: The manuscript was adjusted according to PLOS One style and formatting requirements, including file naming.

3) Please include additional information regarding the survey or questionnaire used in the study and ensure that you have provided sufficient details that others could replicate the analyses. For instance, if you developed a questionnaire as part of this study and it is not under a copyright more restrictive than CC-BY, please include a copy, in both the original language and English, as Supporting Information, or include a citation if it has been published previously.

Response: We thank the Editor for the comment. We used the standardized WHO STEPs survey instrument which extensively utilized for the risk factor surveillance worldwide (reference was added). This instrument was translated and then back-translated. The full information about the questionnaire was added in lines 95-103 in the methods section. We also attached a Russian translated version of the questionnaire to the supplementary materials.

4) In the Methods, please discuss whether and how the questionnaire was validated and/or pre-tested. If these did not occur, please provide the rationale for not doing so.

Response: We acknowledge that information about the questionnaire was not complete. Thus, we added information about the questionnaire in lines 95-103. Please see changes in the main text.

5) In the discussions about your research, please take care to avoid statements implying causality from correlational research, e.g. “iatrogenic factors seem to continue to play a significant role as a route of transmission”. A cross-sectional study can be used to explore associations and not causality due to a number of limitations such a failure to determine temporal precedence.

Response: We appreciate the comment and agree with the suggestion. We rewrote the conclusions. Please see changes in the text.

6) In statistical methods, please clarify whether you corrected for multiple comparisons.

Response: We did not perform multiple pairwise testing among comparison groups. In calculating odds ratios for a categorical variable with three or more groups, we calculated a global p-value for the categorical variable. Thus, there was no need to perform a correction for multiple pairwise testing.

7) In your statistical analyses, please state whether you accounted for survey weights and clustering by region.

Response: We thank the Editor for the valuable suggestion. We recalculated estimates based on the complexity of the study sampling approach. Also, we included the following sentence in the Methods part: “Given that we used two-stage cluster sampling, all descriptive, bivariate and multivariable statistics were weighted using inverse probability weighting to adjust for sample design and non-response rate.” And used multivariable mixed effects logistic regression analysis to account for clustering by region in the study. More details could be found in the statistical analysis part.

8) As part of your revision, please complete and submit a copy of the STROBE checklist, a document that aims to improve reporting and reproducibility of observational studies for purposes of post-publication data analysis and reproducibility: (http://www.strobe-statement.org). Please include your completed checklist as a Supporting Information file. Note that if your paper is accepted for publication, this checklist will be published as part of your article.

Response: The STROBE checklist was completed and attached as Supporting information.

9) We note that the grant information you provided in the ‘Funding Information’ and ‘Financial Disclosure’ sections do not match.

Response: We made corrections in the system so the grant information in the “Funding Information” and “Financial Disclosure” sections match.

10) We note that you have indicated that data from this study are available upon request. PLOS only allows data to be available upon request if there are legal or ethical restrictions on sharing data publicly. For information on unacceptable data access restrictions, please see http://journals.plos.org/plosone/s/data-availability#loc-unacceptable-data-access-restrictions.

Response: The anonymized dataset was provided as Supplementary Information.

Reviewer #1: Overall this is an interesting paper that describes an important disease globally and in Kazakhstan. The paper estimated hepatitis B and hepatitis C for a representative sample of the Kazakhstan population for three large oblasts (provinces). The goal was to relate seroprevalence estimates to several demographic or sociological factors. While important work, there are several major methodological issues that need to be addressed before consideration for publication.

11) ABSTRACT: there are abbreviations for the viruses used in the abstract that need to be spelled out like the authors did in the first paragraph of the introduction.

Response: The abbreviations were spelled out.

12) INTRODUCTION: I'm concerned about calling this a national study when only three oblasts were used to represent a landscape as largest as Kazakhstan. Having been there and worked there for several years, I appreciate that these three areas represent a large proportion of the population. However, I do not think the authors have done enough to describe the distribution of people or describe the proportion of people outside of these three areas and how they may or may not differ from those inside of the study area. I'm not sure it is fair to call this national as designed or at least described.

Response: We thank the Reviewer for this comment. We agree that the study respondents from three regions may not completely represent the general population of Kazakhstan since sex (overrepresented by female participants) and age (the majority were in age group of 40-88 years old) distributions were not comparable to the national demographical statistics. Thus, we modified accordingly the text and the title of the manuscript. This limitation was also mentioned in the discussion part.

13) METHODS: there are several issues that must be addressed prior to consideration for publication. First, as a serological surveillance study I would expect to read a much more clear methodology of how samples were collected, processed, and the serological tests performed. Overall, the methods section lacks appropriate citations and detailed enough methodology to repeat this work. Without expanding on both citations in detail this work should not be considered reproducible.

Response: We thank the reviewer for the comment. We added more details in the methodology on sample collection process and serological testing. Please refer to changes in the study design subsection.

14) In the discussion of the data variables for this study, the authors provide very little information on how urban and rural were defined. These terms are specifically defined in a variety of studies using different cutoffs for population density. Likewise, across these three regions of Kazakhstan, there is a growing periurban environment where rural migrants are moving closer to the city and the city suburbs are expanding into the rural area. This study at a minimum needs to define urban and rural.

Response: We thank the Reviewer for this comment. The threshold for diving into urban and rural areas was 50,000 people in a settlement according to “Law of Administrative-Territorial Structure of Republic of Kazakhstan” (https://online.zakon.kz/m/document/?doc_id=1007265). So, the following definition was added in the methods section: “Any settlement with more than 50,000 people was considered an urban area, while any settlement with at least 50 residents and not exceeding 50,000 people was defined as rural area.”

15) The larger issue with this study is the use of backward stepwise variable selection for the regression modeling. It has been well known for many years now that stepwise selection techniques are losing favor and there is a large body of statistical literature describing why. It has become a best practice to use a multi-model approach, for example, looking at all possible variable combinations with something like a dredge approach. Those models would then be evaluated with something like an AIC. Equally concerning is the use of the spatial location, at least defined by region, as a covariate without first testing for spatial autocorrelation. Each of these three regions was selected because they represent a large proportion of the human population in Kazakhstan. Human populations by their nature are typically clustered in space meaning their covariates are likely autocorrelated. The authors here did nothing to test for spatial autocorrelation either in the input data or in the residuals of the models. Without such tests is difficult to assess whether or not the data for this study meet the assumptions of independence for the regression model selected. Likewise, it has become more usual to see a mixed model approach where a random effect would be used for region. These issues should be addressed and should not simply be addressed by adding one or two sentences to the methodology, but rather performing additional model evaluation, selection, and spatial autocorrelation testing.

Response: We highly appreciate thoughtful comments provided by the Reviewer. We tested for spatial autocorrelation using the Moran’s I test and found dependence of outcomes. Thus, we decided to apply logistic mixed effects regression in statistical analysis. Accordingly, changes were made in the statistical analysis subsection in the methods section. Also, we included the Moran’s I test results in the Supplementary Materials section.

Regression models were built through three stages by including only demographic variables, then additionally adding two risk factors (history of surgery and blood transfusion), and later all other risk factors. This way a reader could compare the models and observe interrelationships of covariates in the models. We felt that building a model based only statistical significance or higher predictive power would lead to not epidemiologically interesting model. That is why, we decided to include epidemiologically important exposure variables (risk factors) despite their statistical non-significance.

16) Building on concerns about the regression model, the authors also used the T-test for the variables without first testing for normality. It may be more likely that a nonparametric test touches the Mann-Whitney U test may be more appropriate to analyze the same question.

Response: We appreciate the comment. Yes, we checked for normality of continuous variables before performing the T-test. Based on the shape distribution of a continuous variable or sufficiently large sample sizes in comparison groups, we selected either the T-test or the Mann-Whitney U test. See changes in the statistical analysis subsection.

17) Later in the results the authors state there were some adjustments for the models and those were not provided.

Response: We added footnotes of description of included variables in the models under each table which presented results from multivariable regression analysis.

18) DISCUSSION: the discussion of the paper is well written. However the discussion highlights the results of the study, which do not address the concerns raised above about the methodology performed. It is important to ensure that the statistical analysis was appropriate before pacing and discussion that supports those results.

Response: After using weighted logistic mixed effects regression analysis, some findings did change. Based on changes in the results, we modified text in the discussion part. Specifically, a history of surgery was not anymore statistically significantly associated with HBsAg seropositivity. Please see the changes in lines 299-314. 

19) I think this is an interesting paper and addresses a very serious disease concern. The methodological comments I raise here can be addressed and should strengthen the paper.

Response: We thank the Reviewer for the valuable comments in improving and strengthening the manuscript. Hope the Reviewer will be satisfied with the provided responses and corresponding changes in the text.

Response: We updated the figures, so they met the PLOS requirements.

---

## [Editor Report · Decision Letter 1]

26 Nov 2021

Seroprevalence and risk factors for hepatitis B and hepatitis C in three large regions of Kazakhstan.

PONE-D-20-31835R1

Dear Dr. Issanov,

We’re pleased to inform you that your manuscript has been judged scientifically suitable for publication and will be formally accepted for publication once it meets all outstanding technical requirements.

Kind regards,

Isabelle Chemin, PhD

Academic Editor

PLOS ONE

Additional Editor Comments (optional):

The answers and improvment after the first round of review were convincing and the paper was improved to reach the desired quality to be published in PlosOne.
---

## [Editor Report · Acceptance letter]

7 Dec 2021

PONE-D-20-31835R1 

Seroprevalence and risk factors for hepatitis B and hepatitis C in three large regions of Kazakhstan. 

Dear Dr. Issanov:

I'm pleased to inform you that your manuscript has been deemed suitable for publication in PLOS ONE. Congratulations! Your manuscript is now with our production department. 

Kind regards, 

on behalf of

Mrs Isabelle Chemin 

Academic Editor

PLOS ONE